

# Diversity and pathogenicity of *Alternaria* species associated with the invasive plant *Ageratina adenophora* and local plants

Yu-Xuan Li[1,*], Xing-Fan Dong[1,*], Ai-Ling Yang[1,2] and Han-Bo Zhang[1]

[1] State Key Laboratory for Conservation and Utilization of Bio-Resources in Yunnan, Yunnan University, Kuming, Yunnan, China

[2] School of Ecology and Environmental Science, Yunnan University, Kuming, Yunnan, China

* These authors contributed equally to this work.

## ABSTRACT

Pathogen accumulation after introduction is unavoidable for exotic plants over a long period of time. Therefore, it is important to understand whether plant invasion promotes novel pathogen emergence and increases the risk of pathogen movement among agricultural, horticultural, and wild native plants. In this study, we used multiple gene analysis to characterize the species composition of 104 isolates of *Alternaria* obtained from the invasive plant *Ageratina adenophora* and native plants from Yunnan, Hubei, Guizhou, Sichuan, and Guangxi in China. Phylogenetically, these strains were from *A. alternata* (88.5%), *A. gossypina* (10.6%) and *A. steviae* (0.9%). There was a high amount of sharing between strains associated with *A. adenophora* and with local plants. Pathogenicity tests indicated that most of these *Alternaria* strains are generalists; the isolates with a wider host range were more virulent to the plant. Woody plants were more resistant to these strains than herbaceous plants and vines. However, the invasive plant *A. adenophora* was highly sensitive to these strains. Our data are valuable for understanding how *A. adenophora* invasion impacts the *Alternaria* species composition of the native plant and whether *A. adenophora* invasion causes potential disease risks in invaded ecosystems.

## INTRODUCTION

Biological invasion has been increasingly viewed as an issue of national security due to its great socioeconomic threats to agriculture, forestry and human health (*Ricciardi, Palmer & Yan, 2011*; *Richardson & Ricciardi, 2013*). Many hypotheses have been developed to explain why invasive plants succeed in introduced ranges (*Jeschke, 2014*), such as the biotic resistance hypothesis (*Levine & D'Antonio, 1999*), evolution of increased competitive ability hypothesis (*Blossey & Notzold, 1995*) and novel weapon hypothesis (*He et al., 2009*). The enemy release hypothesis (ERH) suggests that invasive plants outcompete native species partially due to the lack of specific natural enemies, especially pathogens, in the invaded areas (*Keane & Crawley, 2002*). For example, a previous study of 473 plant species introduced from Europe to the United States showed that 84% fewer fungi and 24% fewer

Corresponding author
Han-Bo Zhang, zhhb@ynu.edu.cn

virus species had infected each plant species in its naturalized range than in its native range on average (*Mitchell & Power, 2003*). One of the reasons for *Silene latifolia* invasion into North America is that two specialists (seed predator and anther smut fungus) occurring in Europe are scarce or lacking in North America (*Wolfe, 2003*). *Halbritter et al. (2011)* found that two specific pathogens for *Brachypodium sylvatcum* were more common in the native range than in the invaded range.

Nonetheless, pathogen accumulation after the introduction of exotic plants is unavoidable. In some cases, pathogen accumulation can hold the spread of invasive plants (*Bohl Stricker et al., 2016*). However, the accumulated pathogens are predicted to affect native susceptible hosts if pathogens transmit in invaded ecosystems. Such dynamics are termed 'spillover' when the pathogens are nonnative and introduced with the invader and 'spillback' when an invasive species hosts native pathogens (*Flory, Clay & Thrall, 2013*). Both processes may indirectly exacerbate the effect of invasions if pathogens reduce the performance and competitive inhibition of co-occurring native species (*Kelly et al., 2009*; *Zhang et al., 2014*). Therefore, the hypothesis of 'accumulation of local pathogens' believes that pathogens accumulated on invasive alien plants may spread to native plants and indirectly enhance the competitive advantage in cases where alien species are more tolerant to pathogens than native plants (*Eppinga et al., 2006*).

On the other hand, these processes may also promote novel pathogen emergence and amplification and increase disease risk in native species. Currently, many examples of the acquisition of a native parasite by exotic species spillbacks and spillovers to natives have been recorded. For example, of the 40 animal nonindigenous species, 70% acquired ≥4 native parasites, and 15% acquired >10 native parasites (*Kelly et al., 2009*). Gray squirrels (*Scurius carolinensus*) from North America threaten the replacement of native red squirrels (*Scurius vulgaris*) in the UK, in part due to the transmission of a parapoxvirus that is lethal to red but not to gray squirrels (*Strauss, White & Boots, 2012*). Nonetheless, these studies have focused on animals, and the invisible threat driven by invasive hosts is expected to be common in wild plant communities in the invaded range but has received less attention.

*Ageratina adenophora* is a perennial herbaceous plant of the Compositae native to Central America and has been introduced into Yunnan Province, China, from Myanmar since the 1940s; currently, *A. adenophora* is distributed in southwestern and central China and is one of the 18 most harmful alien invasive plants in China (*Wang & Wang, 2006*). Previously, *A. adenophora* was reported to host diverse fungal endophytes (*Mei et al., 2013*) and leaf spot pathogenic fungi, such as *Passalora ageratinae* and *Baeodromus eupatorii* (*Sharma Poudel et al., 2019*). In particular, when quantifying the sharing of foliar fungal pathogens by the invasive plant *A. adenophora* and its neighbors, our team found that many *Alternaria* spp. can be isolated from healthy leaves and diseased spots of *A. adenophora*, as well as from diseased spots of native plants; pathogenicity tests further verified that some *Alternaria* strains can cause disease on most native plants (*Chen et al., 2020*). *Alternaria* is widely distributed and commonly occurs as saprophytes, endophytes and pathogens (*Nishikawa & Nakashima, 2020*). More than 95% of *Alternaria* species have a wide range of plant pathogens that can cause a variety of

diseases in many economically important crops or ornamental plants, *e.g.*, early blight in potato and tomato (*Kokaeva et al., 2017*), black spot and leaf spot in wheat (*Vergnes et al., 2006*), and leaf spot in cruciferous (*Al-Lami, You & Barbetti, 2018*), Solanaceae (*Liu et al., 2019*) and Asteraceae (*Wu & Wu, 2018*). Therefore, caution should be taken regarding the possible ecological risk in disease transmission on local plants driven by *A. adenophora* invasion. Addressing this issue depends on determining whether there is a sharing between *Alternaria* strains from invasive plants and from local plants, as well as their pathogenicity and host range.

A previous study indicated that most *Alternaria*, occurring as both endophytes and pathogens on *A. adenophora*, as well as co-occurring local native plants, had the same internal transcribed spacer (ITS) genotype (*Chen et al., 2020*). Because there are few intraspecies and even interspecies variation in the ITS gene for discriminating fungal species (*Yamamoto & Bibby, 2014*), it is necessary to use an analysis of multigene fragments to determine the phylogenetic position of these *Alternaria* strains to judge whether fungal genotypes of *Alternaria* could potentially jump between the invasive plant *A. adenophora* and local host plants. In this study, the phylogenetic positions of *Alternaria* strains isolated from healthy and diseased leaves of *A. adenophora* from Southwest China, as well as diseased leaves of surrounding plants, were determined by Alt a1 and calmodulin gene segments, which are commonly used in the identification of *Alternaria alternate* (*Lawrence et al., 2013*); then, the pathogenicity of these *Alternaria* strains on the invasive plant *A. adenophora*, as well as on native plants, was tested. Our study is valuable for understanding the impact of *A. adenophora* invasion on the *Alternaria* species composition of native plants and the potential disease risks. It can also provide evidence that *Alternaria* can be candidates for the development of biocontrol fungi for *A. adenophora* invasion.

## MATERIALS AND METHODS

### Isolation of fungi

The *Alternaria* strains used in this study were isolated from healthy leaves of *A. adenophora*, diseased leaves of *A. adenophora* and native plants. Leaf samples were collected from Yunnan, Guizhou, Guangxi and Hubei Provinces in China. Some strains from Yunnan were previously reported in our team work (*Chen et al., 2020*). The samples were packed in plastic bags, labeled, and transported to the laboratory. The foliar fungi were isolated and cultured according to the method described by *Arnold & Lutzoni (2007)*. The leaves were rinsed with tap water and then surface sterilized (2% sodium hypochlorite for 30 s and 75% ethanol for 2 min and rinsed with sterile water three times). Healthy leaf tissue or diseased tissue was cut into ~6 mm$^2$ fragments, and then fragments were subsequently plated onto potato dextrose agar (PDA) and cultured in a constant temperature incubator at 28 °C for 3–5 days. When fungi grew out from the tissue segment, hyphal fragments were picked up and transferred to PDA and cultured at 28 °C. All fungi were maintained as pure cultures at Yunnan University (Kunming, China).

## Molecular identification

Fungal genomic DNA was extracted from the isolated fungi according to the method of *Zolan & Pukkila (1986)* and used as a template for PCR. Alt a1 is a specific gene fragment of *Alternaria* spp., which can be used to identify *Alternaria* spp. Therefore, Alt a1 fragments of each isolate were first amplified and sequenced, and Alt-4for and Alt-4rev were used for Alt a1 amplification (Alt-4for; 5′-ATGCAGTTCACCACCATCGCYTC-3′ and Alt-4rev; 5′-ACGAGGGTGAYGTAGGCGTCRG-3′) (*Lawrence, Park & Pryor, 2012*). PCR was performed in a 50 µL reaction volume, which included 1 µL template DNA, 25 µL of 2 × PCR Master Mix (TsingKe, Beijing, China), 1 µL of each forward and reverse primer, and 22 µL of ddH$_2$O. They were subjected to thermal cycling on a gradient PCR machine (Thermo Fisher, Waltham, MA, USA). Amplification products were detected using gel electrophoresis, and the PCR products were sent to the Shanghai Sangon Biotech Company for DNA sequencing. The Alt a1 sequences generated in this study were used as queries to search similar DNA sequences in GenBank of the National Center for Biotechnology Information (NCBI) using the basic local alignment search tool (BLAST). The isolates that were confirmed to be *Alternaria* spp. were then amplified and the calmodulin gene fragment was amplified and sequenced, and primers CALDF1 and CALDR1 (CALDF1; 5′-AGCAAGTCTCCGAGTTCAAGG-3′ and CALDR1; 5′-CTTC TGCATCATCAYCTGGACG-3′) were used for calmodulin amplification (*Lawrence et al., 2013*). All nucleotide sequences generated were used for alignment and correction by SeqMan version 7.0.0 (DNAstar 5.0) and were adjusted and redundant sequences were cut out using BioEdit version 7.0 (*Hall, 1999*). The BLAST function was used to compare the Alt a1 and calmodulin sequence data generated in this study with available sequence data information for type or representative isolates in GenBank of the NCBI (*Al-Lami, You & Barbetti, 2018*). All gene nucleotide sequences reported in this study were deposited at GenBank under the accession numbers OK584830–OK584936 for Alt a-1 and OK584937–OK585043 for calmodulin (also see Supplemental File S1).

## Phylogenetic analysis

These two gene fragments were spliced into a multigene joint dataset in the order of Alt a1-calmodulin. According to previous reports, *Alternaria* spp. sequences of the two gene fragments were downloaded from the GenBank database and were adjusted and cut by the same method described above (*Bertels et al., 2014*). The reference sequence information used is shown in Table 1.

Bayesian inference (BI) and the maximum-likelihood (ML) method were used to construct the phylogenetic tree, and *Alternaria consortialis* (CBS201.67) was used as the outer group for phylogenetic analysis. BI analyses were performed on MrBayes version 3.2.1 (*Ronquist et al., 2012*). jModelTest was used to calculate the most suitable nucleotide substitution model for the experimental data. Metropolis-coupled Markov chain Monte Carlo (MCMCMC) searches were run for 4,000,000 generations, sampling every 100th generation, and until the mean standard deviation of splitting frequency dropped below 0.01. The initial 25% of the generations of MCMCMC sampling were discarded as burn-in. The refinement of the phylogenetic tree was used to estimate BI posterior probability

**Table 1 The reference sequences used for phylogenetic analyses in this study.**

| Species | Source[a] | Locality, host | GenBank accession[b] | |
|---|---|---|---|---|
| | | | Alt a1 | Calmodulin |
| *Alternaria alternata* | CBS 102603 | Israel, *Minneola tangelo* | KP123882 | MH168346 |
| | CBS 106.24 | USA, *Malus sylvestris* | KP123847 | MH168350 |
| | CBS 106.34 | Unknown, *Linum usitatissimum* | KP123853 | JQ646197 |
| | CBS 118811 | USA, *Brassica oleracea* | KP123904 | MH107302 |
| | CBS 118812 | USA, *Daucus carota* | KP123905 | MH175184 |
| | CBS 119543 | USA, *Citrus paradisi* | KP123911 | MH107304 |
| | CBS 121454 | USA, *Cuscuta gronovii* | JQ646402 | MH175186 |
| | CBS 121456 | China, *Sanguisorba officinalis* | KP123917 | MH168352 |
| | CBS 127671 | USA, *Stanleya pinnata* | KP123929 | MH137286 |
| | CBS 194.86 | USA, *Quercus* sp. | KP123869 | MH168351 |
| | CBS 595.93 | Japan, *Pyrus pyrifolia* | JQ646399 | JQ646204 |
| *Alternaria gossypina* | CBS 100.23 | Unknown, *Malus domestica* | KP123977 | JQ646201 |
| | CBS 104.32 | Zimbabwe, *Gossypium* sp. | JQ646395 | JQ646202 |
| *Alternaria steviae* | CBS 632.88 | Japan, *Stevia rebaudiana* | JQ646423 | JQ646240 |
| *Alternaria consortialis* | CBS 201.67 | np[b] | FJ266509 | JQ646173 |

Notes:
[a] CBS, Centraalbureau voor Schimmelcultures, Royal Netherlands Academy of Arts and Sciences, Uppsalalaan 8,3584 CT Utrecht, Netherlands.
[b] np: no product.

values. The tree was viewed in FigTree version 1.4. ML analysis was computed with the PHY files generated with Clustal X version 2.1 (*Thompson et al., 1998*), performed on MEGA X (*Kumar et al., 2018*) and using the GTR-GAMMA model. ML bootstrap proportions were computed with 1,000 replicates.

## Morphological characteristics

According to the phylogenetic tree, the representative isolates were randomly selected for culture on PDA and potato carrot agar (PCA) for observation of conidia and colony morphology. These strains were incubated at 28 °C in a constant temperature incubator for 7 days, and each isolate had five repeats. After 7 days, the diameter of the colony was measured. Then, the isolate was inoculated in V8 Juice by the trisection method and cultured at room temperature for 14 days. The surface of the colonies was gently scraped with cover glass slides and placed on slides dripped with oil and sterilized deionized water for observation of conidia under a microscope. The spore length, width, number of septa and mediastinum, and beaks were measured from 50 conidia that were randomly selected.

## Pathogenicity tests

The pathogenicity of these *Alternaria* strains on *A. adenophora* and native plants was tested by previously described methods (*Gilbert & Webb, 2007*). The field site was located in Xishan Forest Park, Kunming, at an altitude of 2,214 m, latitude of 24°58024″N and longitude of 102°37017″E. Briefly, the selected isolates were drilled with a sterilized

perforator with a diameter of ~6 mm to make a PDA agar disc with fungal mycelium. The mature and healthy leaves of the tested plants were punctured on the underside using a sterile puncher, and the inoculum agar was pressed against the wound on the underside of the leaf using Scotch tape, which was then clipped in place with a bent hair clip. Each isolate was repeatedly inoculated with five leaves, and the PDA agar disc without fungal mycelium was used as a control. Seven days after inoculation, the tested leaves were cut and placed in a sterile plastic bag and transported to the laboratory for observation and measurement. The tested plants included the invasive plant *A. adenophora*, as well as nine local common plants in Kunming, including woody plants: *Cyclobalanopsis glaucoides, Celtis tetrandra*, and *Lindera communis*; herbaceous plants: *Arthraxon hispidu, Hypoestes triflora*, and *Urena lobate*; and vine plants: *Fallopia multiflora, Argyreia pierreana*, and *Ampelopsis bodinieri*.

### Data analysis

One-way ANOVA was used to compare the growth of isolates in different culture media, as well as the pathogenicity of *Alternaria* spp. against *A. adenophora* and native plants. Both Duncan's test and Tukey's test were used for pairwise comparison of the pathogenicity across different groups of *Alternaria* within the same category of plant (*e.g.*, within *A. adenophora* or within native plants). A regression analysis between average spot size and number of hosts was performed to test whether fungal virulence is related to the host range. The calculated relative size based on the pathogenicity was used to show the pathogenicity of *Alternaria* on the invasive plant and local plants using a bubble plot.

## RESULTS

### Fungal isolates and phylogenetic analysis

In total, 104 isolates of *Alternaria* spp. were obtained from *A. adenophora* and native plants from Yunnan (60 isolates), Hubei (17 isolates), Guizhou (18 isolates), Sichuan (8 isolates), and Guangxi (1 isolate). Among them, 32 isolates were from *A. adenophora* (five from healthy leaves and the rest were isolated from leaf spots) and 72 isolates were from leaf spots of native plants (see Table S1 for details).

In the phylogenetic tree of the Alt a1-calmodulin gene, the isolates were divided into four groups: two groups from *A. alternata* (88.5%) and two from *A. gossypina* (10.6%) and *A. steviae* (0.9%) (Fig. 1). Both single-gene phylogenetic trees of Alt a1 and calmodulin showed that isolates belonging to the *A. alternata* section were also divided into two groups (including 92 isolates); however, there were differences in the composition of the isolates in each group (Figs. S1 and S2; Table S1). Regardless of the single- or double-gene tree, the isolates from *A. adenophora* and native plants were grouped together, and many strains showed the same sequence. Interestingly, *Alternaria alternata* was mainly obtained from *A. adenophora*, but those from *A. gossypina* were mainly from native plants (Fig. 1).

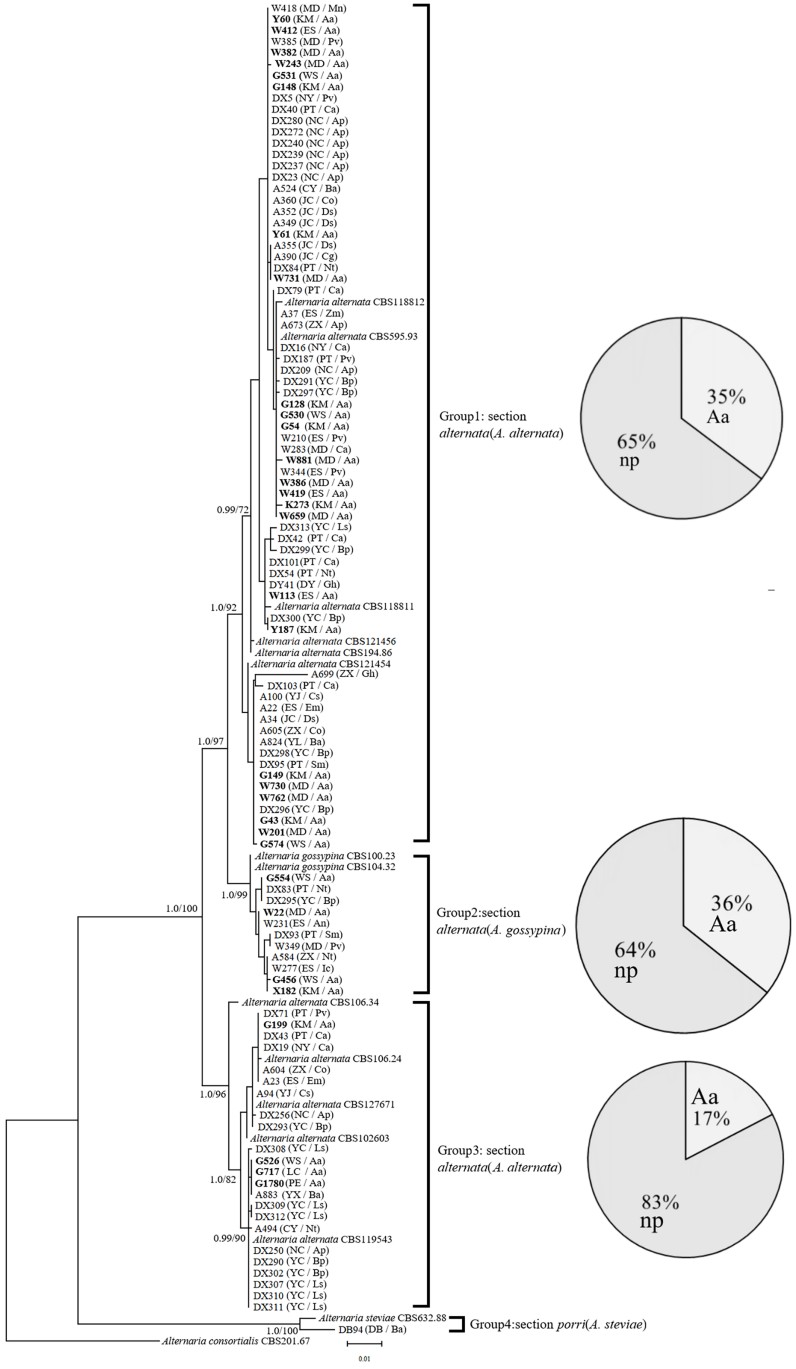

**Figure 1 Phylogenetic tree derived from Bayesian analysis based on combined Alt a1 and Calmodulin sequences of 119 strains representing species in *Alternaria*.** The numbers above branches represent Bayesian posterior probabilities and maximum-likelihood bootstrap percentages (PP/ML). Only bootstrap percentages over 50% and significant Bayesian posterior probability (0.8) are shown on the branches. The geographic location and plant source for each strain are shown in parentheses following the strain number. The numbers in bold are isolates from *A. adenophora*. Geographic location: CY-Cangyuan, DB-Debao, DY-Duyun, ES-Eshan, JC-Jianchuan, KM-Kunming, LC-Lancang, MD-Midu, NC-Nanchong, NY-Nayong, PE-Puer, PT-Pingtang, WS-Weishan, YJ-Yuanjiang, YL-Yiliang, YX-Yunxian, ZX-Zhenxiong; plant source: Aa-*Ageratina adenophora*, An-*Alnus nepalensis*, Ap-*Amygdalus persica*, Ba-*BetuLa alnoides*, Bp-*Brassica pekinensis*, Ca-*Capsicum annuum*, Co-*Cynanchum otophyllum*,

![PeerJ]

**Figure 1** (continued)
Cs-*Camellia sinensis*, Ds-*Dioscorea subcalva*, Em-*Euphorbia milii*, Fm-*Fallopia muLtiflora*, Gh-*Gonostegia hirta*, Ic-*Imperata cylindrica*, Ls-*Lactuca sativa*, Mn-*Musa nana*, Nt-*Nicotiana tabacum*, Pv-*Phaseolus vulgaris*, Ri-*Reinwardtia indica*, Sm-*Solanum melongena*, Zm-*Zehneria maysorensis*. The tree was rooted to *A. consortialis* (CBS201.67). The right side of each group shows the percentage of strains isolated from invasive plants (Aa, *A. adenophora*) and native plants (np, native plant) in this group.

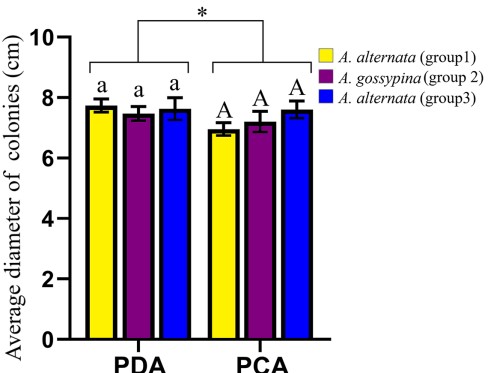

**Figure 2 Growth diameter of isolates on different culture media.** The same letter indicates that there is no significant difference for different groups on PDA or PCA. One-way ANOVA was used to compare the growth of isolates in different culture media ($F = 4.070$, $P = 0.055$). Identical lowercase or uppercase letters indicate nonsignificant differences. "*" indicates marginally significant ($P < 0.10$).

## Morphological analysis

The representative morphology of conidia and colonies for these *Alternaria* strains are shown in Fig. S3. The conidia were brown to black and inverted rod-shaped, ovoid or nearly elliptical, with 3–6 transverse septa and 0–3 longitudinal septa, and were always beakless or pseudorostrate. Whether on PDA or PCA, the colonies were round and fluffy, without pigment production, with the exception of DB94 (belonging to *A. steviae*), which produced orange pigments. The colony color varied greatly among strains between groups, as well as within groups. The colony diameters on different media were marginally different, but there was no difference on the same media for different groups (Fig. 2).

## Pathogenicity analysis

In total, 52 isolates were randomly selected to test pathogenicity on *A. adenophora* and native plants. For *A. adenophora*, 35 of 42 tested *A. alternata* strains were pathogenic, without a difference between those from Groups 1 and 3; and 4 of 9 tested isolates belonging to *A. gossypina* were pathogenic (Fig. 3). In general, *A. alternata* strains (particularly Group 3) were more virulent than *A. gossypina* (Fig. 4A; Table 2).

For nine tested native plants, most of these *Alternaria* strains are generalists, and each isolate was pathogenic to at least one native plant. Only three isolates were pathogenic to only one native plant (Fig. 3). The plant *Hypoestes triflora* was the most sensitive host,

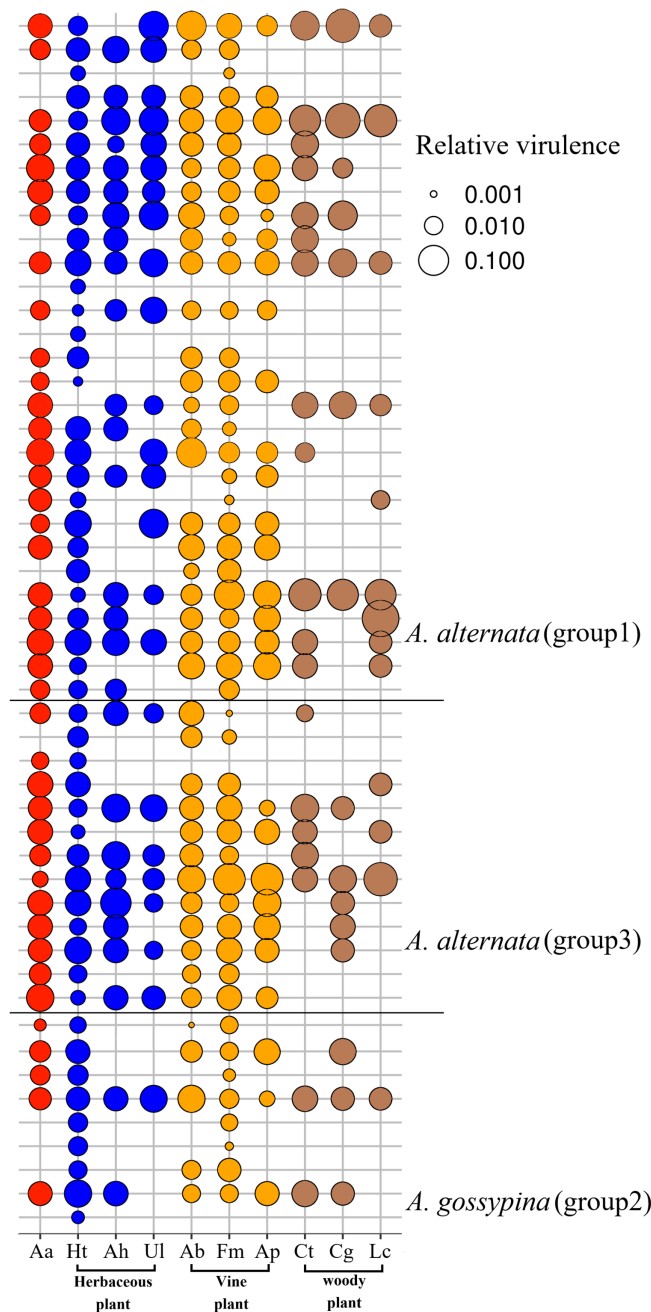

**Figure 3 Bubble plot for the pathogenicity of *Alternaria* on the invasive plant and local plants.** Aa, *A. adenophora*, Ht, *Hypoestes trifloral*, Ah, *Arthraxon hispidu*, Ul, *Urena lobata*, Ab, *Ampelopsis bodinieri*, Ap, *Argyreia pierreana*, Fm, *Fallopia multiflora*, Ct, *Celtis tetrandra*, Cg, *Cyclobalanopsis glaucoides*, Lc, *Lindera communis*. The bubble area is the calculated relative size based on the pathogenicity.

resisting only one isolate, while *Lindera communis* was the least sensitive host, resisting 38 isolates (Fig. 3). In general, the isolates with a wider host range were more virulent to the plant (Fig. 4C) (see Table S3 for the spot area data after infection).

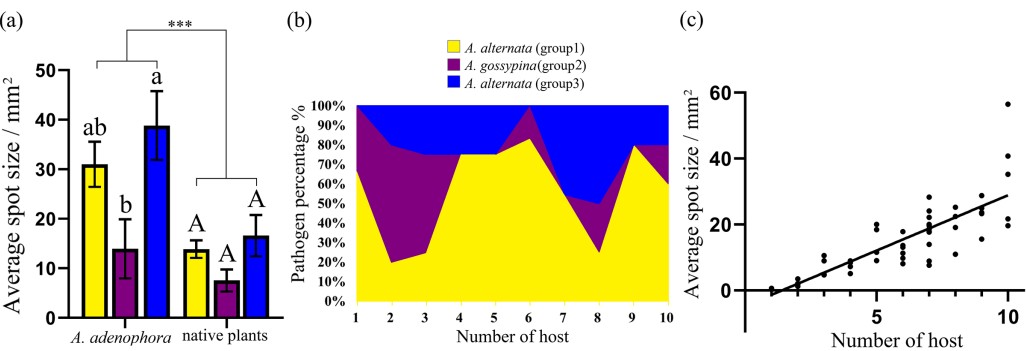

**Figure 4 Comparison of the pathogenicity (A) and host range (B) of three groups of *Alternaria* spp. on *A. adenophora* and native plants and correlation analysis of host range and pathogenicity (C).** The error bar represents the standard error. One-way ANOVA was used to compare the pathogenicity of *Alternaria* spp. against *A. adenophora* and native plants ($F = 18.940$, $P < 0.001$), and Duncan's test and Tukey's test were used for pairwise comparison of different groups of *Alternaria* within the same category of plants (*e.g.*, within *A. adenophora* or within native plants). This figure shows the results of Tukey's test. Different lowercase letters indicate that the difference was significant, and identical lowercase or uppercase letters indicate nonsignificant differences. Asterisks (***) indicates extremely significant ($P < 0.001$). The equation for the regression of average spot size and number of hosts was $Y = 3.312 * X - 4.578$ ($R^2 = 0.677$, $P < 0.001$).

**Table 2 Results of one-way ANOVA with different test methods.**

| Genotype | Average spot size/mm² (Aa)[a] | | Average spot size/mm² (np)[a] | |
|---|---|---|---|---|
| | Duncan | Tuckey | Duncan | Tuckey |
| A. a (group1) | 31.00 ± 24.51ab | 31.00 ± 24.51ab | 13.87 ± 8.82AB | 13.87 ± 8.82A |
| A. g (group2) | 13.93 ± 17.83b | 13.93 ± 17.8b | 7.55 ± 6.68B | 7.55 ± 6.68A |
| A. a (group3) | 38.83 ± 25.00a | 38.83 ± 25.00a | 16.62 ± 15.02A | 16.62 ± 15.02A |

**Note:**
[a]Aa, *A. adenophora*; np, native plant.

## DISCUSSION

Our study is the first to determine the phylogenetics and pathogenicity of *Alternaria* associated with an invasive plant and native plants. In total, 104 *Alternaria* strains were divided into four groups, phylogenetically belonging to *A. alternata*, *A. gossypina*, and *A. steviae* (Fig. 1), using previously described genes in the identification of *Alternaria*, including *Alternaria* major allergen (ALT) and calmodulin (*Lawrence et al., 2013*). Some strains belonging to *Alternaria alternata* were different in the calmodulin and Alt a1 phylogenetic trees (Fig. 1; Figs. S1 and S2), suggesting that the section *Alternaria alternata* harbors more diverse genetic variation than *A. gossypina*.

Again, our multiple gene analysis indicated that there was still great sharing between the isolates from *A. adenophora* and from native plants (Fig. 1), supporting a previous report revealed by ITS gene (*Chen et al., 2020*). Such a sharing indirectly suggests a high possibility for these *Alternaria* of host jumps between invasive plants and surrounding native plants. This is common for fungal pathogens of hosts to jump (*Silva et al., 2012*; *Slippers, Stenlid & Wingfield, 2005*). As an invading host becomes more abundant in the community, it can increase the frequency of those pathogen genotypes most able to infect

and reproduce on the dominant host species (*Gilbert & Parker, 2010*). For example, the ability of fungal generalists to undergo range expansion is probably due to their capacity to infect novel hosts (*Brown & Hovmøller, 2002*; *Evangelista et al., 2008*). Nonetheless, whether these *Alternaria* strains exhibit host jumps requires further evidence, including the dynamics of these *Alternaria* on *A. adenophora* since their introduction in Yunnan, as well as a comparison of *Alternaria* isolated from *A. adenophora* in its native and invaded ranges. Interestingly, most of our strains (~88%) are from Section *Alternaria alternata* (Fig. 1), which is well known to be widely distributed and an important pathogen for many plant species (*Woudenberg et al., 2015*). Therefore, there is a great ecological risk in disease transmission on local plants driven by *A. adenophora* invasion if these *Alternaria* can cause disease in co-occurring local plants.

Indeed, our pathogenicity test further verified that most strains of *A. alternata* are not only virulent to *A. adenophora* but also commonly to native plants (Fig. 3). Therefore, the disease risk to neighboring native plants caused by these shared *Alternaria* fungi should be met with caution. Relative to native plants, invasive exotic species often grow monocultures, are high-density, are poorly defended (*Blumenthal, 2006*; *van Kleunen, Weber & Fischer, 2009*) and are expected to be ideal pathogen reservoirs (*Cronin et al., 2010*). Recently, several examples have been examined in the context of wild plant communities. For example, spillover of barley yellow mosaic virus from a highly susceptible invasive grass decreased the abundance of two native grasses through pathogen-mediated apparent competition (*Power & Mitchell, 2004*). Invasive cheatgrass (*Bromus tectorum*) serves as a reservoir for the native seed pathogen *Pyrenophora semeniperda*, which causes significantly greater death of native seeds in invaded areas (*Beckstead et al., 2010*). In the UK, the invasive *Rhododendron ponticum* is a key foliar reservoir host for both *Phytophthora ramorum* and *P. kernoviae* (*Purse et al., 2012*). Thus, it can be expected that diverse *Alternaria* associated with *A. adenophora* may be potential pathogen sources for co-occurring local plants in the invaded ecosystem. Our current pathogenicity test was performed only in one geographic location under natural conditions (see 'Materials and Methods'). The *Alternaria* spp. isolates in this study were collected from a wide range of geographic locations; thus, caution should be taken when explaining the pathogenicity of *Alternaria* spp. isolates because pathogen virulence varies with environmental conditions such as temperature and humidity.

The hypothesis of 'accumulation of local pathogens' indicates that pathogens accumulated on invasive alien plants may spread to native plants and produce a disadvantage in competition with alien species when alien species are more tolerant to pathogens (*Eppinga et al., 2006*). For example, the invasive *Chromolaena odorata* can accumulate high concentrations of the generalist soil-borne fungal pathogen *Fusarium semitectum* in their invaded range, thereby creating a negative response in native plant species (*Mangla & Callaway, 2007*). However, both species and abundance of pathogens accumulated by invasive plants are highly dynamic along with the expansion range and time (*Mitchell et al., 2010*), it is difficult to evaluate the realized impacts of a given pathogen on introduced host population. In this case, our results showed that such an indirect advantage is a low possible event for *A. adenophora* over native plants through these

*Alternaria* species because *A. alternata* in general is more virulent to *A. adenophora* than to native plants (Fig. 3). Therefore, it is not possible for the disease-mediated invasion of *A. adenophora* by *Alternaria* to act as 'biological weapons' from invaders (*Strauss, White & Boots, 2012*). Nonetheless, whether these *Alternaira* strains can act as 'biological weapons' from invaders depends on which local competitors are selected for evaluation. For example, woody plants, *e.g.*, *Lindera communis* was more resistant to these fungi than the other plants; in particular, the herbaceous plant *H. triflora* was sensitive to 51 strains (Fig. 3). It is therefore expected that *H. triflora* has a disadvantage when competing with *A. adenophora* due to a disease weapon (*Alternaria*).

## CONCLUSIONS

Our study verifies that abundant fungi belonging to *A. alternata*, *A. gossypina* and *A. steviae* inhabit the healthy and diseased leaves of *A. adenophora*, as well as diseased leaves of surrounding local plants. Pathogenicity tests indicated that these *Alternaria* species are generalists and are virulent to *A. adenophora* and common native plants. Therefore, *Alternaria* associated with *A. adenophora* can be a potential disease source for local native plants. Nonetheless, *A. alternata* can cause leaf spot and other diseases in a variety of crops (*Gao et al., 2020*; *Kgatle et al., 2018*). The spillback of these *Alternaira* strains and potential risk to crops remain to be verified. In addition, previous efforts have attempted to develop *Alternaira* as a biocontrol method on *A. adenophora* (*Zhou et al., 2010*). Our data indicated that *Alternaria* with more virulence commonly had a wider range of hosts (Fig. 4). Therefore, it is nearly impossible to obtain a biocontrol strain of *Alternaria alternata* with high virulence and host specificity unless genetic modification is used.

## ACKNOWLEDGEMENTS

The authors thank Tian Zeng, Jie Zhou, Lin Chen, and Kai Fang at Yunnan University for help with sampling in the field and performing the disease experiment. Dr Huan-Chong Wang and Dr Tao Xu at Yunnan University assisted with the identification of plant species.

### Funding

This work was supported by the Major Science and Technology Project in Yunnan Province, China (No. K264202011020) and the National Natural Science Foundation of China (Nos. 31770585 and 31360153). The funders had no role in study design, data collection and analysis, decision to publish, or preparation of the manuscript.

### Grant Disclosures

The following grant information was disclosed by the authors:
Major Science and Technology Project in Yunnan Province, China: K264202011020.
National Natural Science Foundation of China: 31770585 and 31360153.

## Competing Interests

The authors declare that they have no competing interests.

## Author Contributions

- Yu-Xuan Li performed the experiments, analyzed the data, prepared figures and/or tables, authored or reviewed drafts of the paper, and approved the final draft.
- Xing-Fan Dong performed the experiments, analyzed the data, prepared figures and/or tables, and approved the final draft.
- Ai-Ling Yang performed the experiments, authored or reviewed drafts of the paper, and approved the final draft.
- Han-Bo Zhang conceived and designed the experiments, performed the experiments, analyzed the data, authored or reviewed drafts of the paper, and approved the final draft.

## Field Study Permissions

The following information was supplied relating to field study approvals (*i.e.*, approving body and any reference numbers):

Ethics committee of my university.

## DNA Deposition

The following information was supplied regarding the deposition of DNA sequences:

All gene nucleotide sequences reported in this study are available at GenBank: OK584830–OK584936 for Alt a-1 and OK584937–OK585043 for Calmodulin.

## Data Availability

The raw measurements are available in the Supplemental File.

## Supplemental Information

Supplemental information for this article can be found online at http://dx.doi.org/10.7717/peerj.13012#supplemental-information.

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
