# Peer review of "Diversity and pathogenicity of Alternaria species associated with the invasive plant Ageratina adenophora and local plants"

_PeerJ, doi:10.7717/peerj.13012_

## Round 0.1 · original submission · Major Revisions

Dear Dr. Zhang:

I have conditionally accepted your manuscript with MAJOR revision because I believe that the paper is of significant value but will require substantial rewriting before acceptance of the manuscript for publication in PeerJ. Based on the reviewers and my assessment of the manuscript, I am suggesting that you revise your manuscript to address the concerns of the reviewers.

Both reviewers have serious concerns about the experimental designs, data collection, interpretation, and biological significance of your research. Reviewer 1 main concern is that the authors failed to evaluate the strains using robust methods. In this regard, the author must provide the relevant justification(s). Reviewer 2 felt that “the authors are inconclusive concerning the identity of A. gossypina/ longipes, thus, sequencing other genes/genomic regions (GAPDH, OPA10-2, KOG1058) as suggested by Woudenberg et al. (2015) is required”. The manuscript needs a fair bit of language and grammatical editing. I also encourage you to get a proficient English-speaking professional person to help with the revision.

Best regards,

Tika Adhikari

Reviewer 1 ·

Basic reporting

A bit more attention should be paid to the accuracy of sentences. The authors may choose to get the manuscript read by an external expert. Literature citation is adequate. Structure of article and hypothesis behind are OK.

Experimental design

Manuscript is within the scope of the journal. Research question can be better defined at the end of introduction. Methods are described and technically upto the standard. But, please refer to the detailed comments and address them.

Validity of the findings

Suggestions are provided at #4. Overall the rationale, adequacy of data and drawing conclusions are fine to me. A detailed response from the authors are needed if revision is submitted.

Additional comments

The study by Li et al. showed diversity and pathogenicity of Alternaria species associated with the invasive plant Ageratina adenophora and native plants. Li et al. used multiple gene analysis/phylogenetic analysis to characterize the species composition of 107 isolates of Alternaria obtained from the invasive plant Ageratina adenophora and native plants from different provinces of China. Overall, the manuscript is well written, results supports the conclusion and the data presented in this report would be of interest to many readers.
Minor comments
Lines 62-64 Statement is unclear to me. The fitness of the invasive species is entirely depending on the following aspects: (I) the absence of competition and II) their responses in presence of different densities of intra- and inter-specific competitors, therefore, more references are needed to support the argument.
Lines 79-80 Few more examples with appropriate references are needed.
Lines 90-91 Alternaria spp. can also behave as asymptomatic endophytes in invasive species and also cause visible symptoms on co-occurring native plants.
Therefore, what kind of pathomorphological characteristics and disease diagnostic techniques have been employed?
How many days have been taken into account to determine plants are asymptomatic (absence of visible symptoms)?
Line 142 A detail regarding primer sequences should be mentioned.
Lines 187 and 229 Manuscript requires statistical analysis (PCA, Tuckey test for comparing different variables) to support conclusions derived from Figure 3 and Figure 4.
Some robust statistical analysis (PCA, Tuckey tests etc.) should be carried out to show convincingly the relationship between the degree of pathogen virulence and plant population (native/invasive). A table should also include that define the comparative analysis of parameters tested (percentage area with disease symptoms, relative virulence/abundance of fungal isolates/pathogens etc.) across plant species (invasive/local) tested.
Lines 206-208 In addition to geographical location, pathogen susceptibility and resistance strongly varied with environmental conditions such as temperature, heat, drought, salinity and humidity. Do susceptibility of the isolates of Alternaria spp. retrieved from A. adenophora varied with the geographical locations and/or under different environmental gradient?
Lines 230-235 For more clarity, a figure (main/supplemental) using appropriate statistics should be included to show the disease scoring such as infection area (in cm2) of native and invasive plant species inoculated with different A. alternata and A. gossypina/ longipes pathogen species.
Line 260 Host jump is an interesting phenomenon during the process of introduction of invasive plants to a particular geographical location. The authors are requested to add few sentences in the introduction and discussion section. Is it possible to determine host-jump phenomenon on community basis by comparing fungal pathogens/endophytes isolated from invasive plants in its native and invaded ranges?
Lines 289-291 Wide variations related to competitive ability, disease susceptibility and reduced growth and development have been observed in native plant species. Therefore, it is assumed that the difference in response of native plants to fungal infection and competition with an exotic/invasive plant species may suggest the differential response of native plant species to disease-mediated competition. Please elaborate more on this citing appropriate references.

·

Basic reporting

No comment

Experimental design

For accurate taxonomic identification of isolates suspected to be belonging to the Alternaria spp complex, authors are advised to refer to Table 4 of Woudenberg et al. (2015) (title: Alternaria section Alternaria: Species, formae speciales orpathotypes). In this Manuscript, the authors are inconclusive with respect to the identity of A. gossypina/ longipes, therefore, sequencing other genes/genomic regions (gapdh, OPA10-2, KOG1058) as suggested by Woudenberg et al. (2015) is required.

Validity of the findings

No comments

---

## Round 0.2 · accepted · Accept

Dear Dr. Zhang,
Thank you for your submission to PeerJ.

I am writing to inform you that your manuscript - Diversity and pathogenicity of Alternaria species associated with the invasive plant Ageratina adenophora and local plants - has been Accepted for publication.

Congratulations!

Reviewer 1 ·

Basic reporting

See my previous report

Experimental design

See my previous report

Validity of the findings

See my previous report

Additional comments

Comments are addressed.

·

Basic reporting

No comments

Experimental design

Acceptable

Validity of the findings

Acceptable

Additional comments

The manuscript can be accepted in its present form.